## META-RESEARCH

# COVID-19 medical papers have fewer women first authors than expected

**Abstract** The COVID-19 pandemic has resulted in school closures and distancing requirements that have disrupted both work and family life for many. Concerns exist that these disruptions caused by the pandemic may not have influenced men and women researchers equally. Many medical journals have published papers on the pandemic, which were generated by researchers facing the challenges of these disruptions. Here we report the results of an analysis that compared the gender distribution of authors on 1893 medical papers related to the pandemic with that on papers published in the same journals in 2019, for papers with first authors and last authors from the United States. Using mixed-effects regression models, we estimated that the proportion of COVID-19 papers with a woman first author was 19% lower than that for papers published in the same journals in 2019, while our comparisons for last authors and overall proportion of women authors per paper were inconclusive. A closer examination suggested that women's representation as first authors of COVID-19 research was particularly low for papers published in March and April 2020. Our findings are consistent with the idea that the research productivity of women, especially early-career women, has been affected more than the research productivity of men.

**JENS PETER ANDERSEN, MATHIAS WULLUM NIELSEN, NICOLE L SIMONE, RESA E LEWISS AND RESHMA JAGSI***

## Introduction

During the COVID-19 pandemic, many governments have shuttered schools and implemented social distancing requirements that limit options for childcare, while simultaneously requiring researchers to work from home (*Minello, 2020*). Robust evidence suggests that women in academic medicine shoulder more of the burden of domestic labor within their households than do men. One study of an elite sample of NIH-funded physician-researchers showed that women spent 8.5 hr more per week on parenting and domestic tasks than their men peers (*Jolly et al., 2014*). Recent research also suggests that women in academia take on more domestic responsibilities than men, even in dual-career academic couples (*Derrick et al., 2019*). Therefore, the recent restrictions in access to childcare might reasonably be expected to have disproportionate impact on women in academic medicine, as compared to men (*Viglione, 2020*). The impact of new professional service demands that now compete with time for scholarly productivity in academic medicine, including work to increase the use of virtual platforms for teaching and clinical care, may also disproportionately impact women medical researchers, who are disproportionately represented on clinician-educator tracks (*Mayer et al., 2014*).

Here, we focus on the published medical research literature, where it may be possible to provide an early evaluation of whether the gender gap in academic productivity is widening. The medical literature now includes a substantial number of articles directly relating to COVID-19, mostly generated rapidly after the broader social restrictions came into being, in most US states, in March 2020. We identified 15,839 articles on COVID-19 published between 1 January 2020 and 5 June 2020, including 1893

***For correspondence:** rjagsi@
med.umich.edu

**Reviewing editor:** Peter
Rodgers, eLife, United Kingdom

**Figure 1.** COVID-19 papers have fewer female authors than papers from 2019 published in the same journals. (a–c) Observed (bars) and estimated (crosses and error-bars) proportions of women among authors of 1,893 US papers on COVID-19 and 85,373 papers published in the same journals in 2019. The bars show differences in the observed proportions of women in the first-author position (a), the last-author position (b), and any author position (c), for papers published in 2020 COVID-19 papers (blue bars) versus papers from the same journals in 2019 (orange bars). All three panels suggest a decrease in the observed proportion of women. The crosses and error bars show the adjusted means and 95% confidence intervals (CIs) derived from mixed regression models with scientific journal as random effect parameter. (d–f) Adjusted means (crosses) and 95% CIs (error bars) derived from mixed regression models for the proportion of women in the first-author position (d), last-author position (e) and any author position (f), for papers published in 2019 (left-most crosses and error bars in each panel), papers published in March and April 2020 (middle), and papers published in May 2020 (right). For all models, there is a drop in March and April, followed by a partial resurgence in May. However, the uncertainty of the estimates make these comparisons inconclusive. See *Supplementary file 1* for details of the mixed regression models used to estimate adjusted means and 95% CIs.

articles that had a first author and/or last author with an affiliation in the US. Here we report the results of an analysis that compared the proportion of women scientists in various author positions in this sample and a sample of 85,373 papers published in the same journals in 2019

(with first and/or last authors with a US affiliation; see Materials and methods for details).

## Results

In *Figure 1a–c* we juxtapose the observed proportion of women authors (bars) for COVID-19

papers and for papers published in the same journals in 2019. This descriptive analysis suggests that women's respective share of first authorships (panel a), last authorships (panel b) and overall representation per paper (panel c) is 14%, 3% and 5% lower for COVID-19 papers compared to 2019 papers (COVID-19 sample: first authorships, arithmetic mean = 0.33; last authorships, arithmetic mean: 0.28, overall proportion: 0.33; 2019 sample: first authorships, arithmetic mean = 0.38; last authorships, arithmetic mean: 0.29; overall proportion: arithmetic mean = 0.35).

The crosses and error-bars in *Figure 1a–c* plot the adjusted means and 95% confidence intervals derived from three mixed regression models that adjust for variations in COVID-19 related research activities across scientific journals. The plots suggest that women's estimated share of first authorships, last authorships, and overall proportion per paper is 19%, 5% and 8% lower in the COVID-19 sample (first authorships, adjusted mean = 0.32, CI: 0.28–0.36; last authorships, adjusted mean: 0.26, CI: 0.23–0.30; overall proportion, adjusted mean = 0.36, CI: 0.33–0.30) than in the 2019 sample (first authorships, adjusted mean: 0.40, CI: 0.37–0.42; last authorships, adjusted mean: 0.28, CI: 0.26–0.31; overall proportion, adjusted mean: 0.38, CI: 0.36–0.40) (see *Supplementary file 1* for model specifications). However, as indicated by the overlapping confidence intervals in panels b and c, the results are inconclusive for last authorships and for the overall proportion of women per paper.

An earlier iteration of this study (https://arxiv.org/abs/2005.06303v2) based on COVID-19 papers published between 1 January 2020 and 5 May 2020 suggested larger differences than those reported here. Specifically, we found that women's share of first authorships, last authorships and general representation per author group was 23%, 16% and 16% lower for COVID-19 papers compared to 2019 papers published in the same journals. The present analysis covers a larger publication window of COVID-19 research (between 1 January 2020 and 5 June 2020), which has increased the sample from 1,179 US-based COVID-19 papers to 1893 (61% increase). Moreover, the present analysis is restricted to COVID-19 papers authored by US-based first and/or last authors, while the prior analysis included all papers with at least one US-based author. While this difference in sampling criteria might explain part of the observed variation in outcomes, we wanted to examine whether there has been a change over time. In

*Figure 1d–f* we report the estimated proportion of women first authors (panel d), last authors (panel e) and overall representation per paper (panel f), for studies published in 2019 (orange crosshairs), during March and April 2020 (blue crosshairs) and in May 2020 (purple crosshairs). All three models indicate lower participation rates for women in March and April 2020 compared to May 2020, but the uncertainty of the estimates make these results inconclusive. However, panel d shows that the relative difference between women's proportion of first-authored COVID-19 papers compared to 2019 papers increases to 23%, when the COVID-19 sample is restricted to papers published in March and April 2020.

To obtain a closer approximation of differences across research areas, we calculated the proportion of women authorships per journal specialty. As shown in *Table 1*, women are represented at lower rates across most specialty groupings in the COVID-19 sample as compared to the 2019 sample. The relative gap in women's participation is most salient in infectious diseases, radiology, pathology, and public health. Importantly, none of these groups show extreme deviations from the overall trend. This suggests that the observed differences are not due to a journal-specialty bias, where specialties with a high representation of men produce the majority of COVID-19 research.

## Discussion

Prior research has raised concerns about women's underrepresentation among authors of medical research, including both original research and commentaries (*Clark et al., 2017*; *Hart and Perlis, 2019*; *Jagsi et al., 2006*; *Larson et al., 2019*; *Silver et al., 2018*). Our study suggests that the COVID-19 pandemic might have amplified this gender gap in the medical literature. Specifically, we find that women constitute a lower share of first authors of articles on COVID-19, as compared to the proportion of women among first authors of all articles published in the same journals the previous year. However, our analysis also indicates that the first-author gender gap in COVID-19 research might have decreased during the past month of the pandemic. Our findings are consistent with a contemporaneous study of pre-prints (*Vincent-Lamarre et al., 2020*), which also found women to be under-represented.

Our findings are consistent with the idea that restricted access to child-care and increased

**Table 1.** Proportion of women authors on 2019 papers and COVID-19 papers by specialty.

Number of observations, N, and proportion of women by author list position for journals grouped by their specialty. The grouped columns show results by journal specialty for COVID papers published in 2020 (four rightmost columns) in contrast to papers from the same journals in 2019. Only papers with a US-based first and/or last author and clear gender for first and last author are included.

| | | 2019 papers | | | | COVID-19 papers | | |
| --- | --- | --- | --- | --- | --- | --- | --- | --- |
| | | Proportion of women | | | | Proportion of women | | |
| *Journal specialty* | N | *First author* | *Full group* | *Last author* | N | *First author* | *Full group* | *Last author* |
| Dermatology | 1811 | 0.44 | 0.42 | 0.37 | 72 | 0.46 | 0.41 | 0.31 |
| Emergency medicine | 1283 | 0.32 | 0.30 | 0.22 | 54 | 0.31 | 0.25 | 0.13 |
| High impact general medicine | 7142 | 0.41 | 0.42 | 0.39 | 194 | 0.31 | 0.37 | 0.35 |
| Infectious diseases | 1404 | 0.45 | 0.42 | 0.34 | 44 | 0.20 | 0.32 | 0.34 |
| Internal medicine | 19,980 | 0.36 | 0.33 | 0.25 | 484 | 0.33 | 0.32 | 0.24 |
| Other basic sciences | 6975 | 0.42 | 0.38 | 0.29 | 135 | 0.33 | 0.34 | 0.28 |
| Other clinical sciences | 21,869 | 0.40 | 0.37 | 0.31 | 429 | 0.38 | 0.38 | 0.35 |
| Otolaryngology | 1063 | 0.32 | 0.29 | 0.21 | 106 | 0.28 | 0.29 | 0.24 |
| Pathology | 869 | 0.46 | 0.43 | 0.32 | 66 | 0.27 | 0.37 | 0.30 |
| Public health | 11,015 | 0.47 | 0.41 | 0.35 | 99 | 0.33 | 0.41 | 0.37 |
| Radiology | 2262 | 0.37 | 0.33 | 0.27 | 60 | 0.25 | 0.28 | 0.17 |
| Surgery | 9700 | 0.21 | 0.20 | 0.13 | 186 | 0.26 | 0.22 | 0.16 |

work-related service demands might have taken the greatest toll on early-career women, particularly early on when the disruptions were most unexpected, although our observational data cannot conclusively support causal claims. As more robust evidence becomes available, mechanisms which disadvantage specific ethnic, age and gender groups should be monitored and inform policies that promote equity (*Donald, 2020*).

Some have argued that the authorship gender gap in academic medicine is best explained by a slow pipeline and the historical exclusion of women from medical school enrollment (*Association of American Medical Colleges, 2019*). However, as time has passed, and women have reached parity in the United States and even begun to constitute the majority of the medical student body in many other countries, their persistently low participation as authors has raised concerns about bias in unblinded peer review processes and unequal opportunities prior to manuscript submission (*Jagsi et al., 2014*; *Silver, 2019*). Studies have demonstrated differences in the language used by men and women to describe their research findings (*Lerchenmueller et al., 2019*), and evidence from the field of economics suggests that women's writing may be held to higher standards (*Hengel, 2017*). In any case, the current study suggest that if authorship of COVID-19-related papers is a bellwether, women's participation in the medical research literature may now be facing even greater challenges than before the pandemic (*Kissler et al., 2020*).

This study is limited to a relatively small sample produced early in the course of the pandemic and misses information on important covariates. A key limitation is that we have not been able to adjust for variations in COVID-19 related research activities across medical research specialties. Since women's representation as authors varies across specialties (*Andersen et al., 2019*), this may introduce a bias. We have attempted to mitigate this bias by including scientific journal as random effect parameter in the regression models, hereby adjusting for variations in COVID-19 related research activities across publication outlets. Moreover, descriptive analysis that breaks down our results by journal specialty does not suggest that those journal specialties that might dominate research related to COVID had low proportions of women among authors in 2019. Indeed, many such specialties, including infectious disease and public health, qualitatively appear to have a markedly lower proportion of women among authors in the 2020 COVID-related dataset than in the 2019 dataset within those fields. Nevertheless, future research might refine our analysis by using Medical Subject Headings (MeSH) to infer the research specialty of each

paper (*Andersen et al., 2019*). The US National Library of Medicine usually assign MeSH terms to medical papers within 3–6 months after publication.

Although we were reliably able to determine gender for the vast majority of the first and last authors and a large majority of all authors, bias is possible due to omission of those whose gender could not be determined. There is no difference in the percentage of matched names between the treatment and control groups.

Despite limitations, this early look suggests that the previously documented gender gap in academic medical publishing may warrant renewed attention (*Jagsi et al., 2006*), and that ongoing research on this subject is necessary as more data become available. The need for greater equity and diversity is most evident in times of crisis. Abundant literature reveals the importance of diverse teams for solving complex problems like those related to COVID-19 (*Mayer et al., 2014*; *Nielsen et al., 2017a*; *Nielsen et al., 2018*; *Phillips, 2014*; *Woolley et al., 2010*). If societal constraints limit the talent pool who may contribute to research informing the crisis response, the consequences will be profound indeed. Policies to support the full inclusion of diverse scholars and transformation of norms for dividing labor appear to be urgent priorities. Policies that merit consideration include providing more teaching support for female faculty or relieving them of teaching duties, supporting child-care costs and identifying child-care options, extending the tenure-clock for the duration of the lockdown, or adjusting the criteria used to assess and select candidates for research funding and tenured positions.

## Materials and methods

On 5 June 2020 we searched PubMed Medline for papers including 'COVID-19' or 'SARS-CoV2' in the title or abstract, to identify publications most likely generated after pandemic-related societal changes developed. This resulted in 15,843 articles, of which only four were published prior to 2020. We extracted journal information and matched the 2020 papers [treatment] to 2019 papers [control] from the same journals ($N$ = 316,367). Only journals with at least five papers on COVID-19 were included in the analysis (629 of 2420 journals (25.9%), 12,855 of 15,843 papers (81.1%)). We extracted author names for both treatment and control, and used these to determine author gender as

in prior work (*Andersen et al., 2019*; *Nielsen et al., 2017b*). Please see these papers for a clarification of the gender-API algorithm and our robustness checks of gender inference.

Gender was reliably estimated for 90.2% of the entire sample. The majority of insecure inferences are due to Chinese names, which are commonly not gendered (*Andersen et al., 2019*; *Nielsen et al., 2017b*). For the papers with at least one US author, gender could be established for 90.7% of US first authors and 91.7% of US last authors. Only papers with gender reliably identified for first and last authors were included. Limiting the sample further to papers with at first author and/or last author with a US address, with gender determined for authors, gives us a treatment group of 1893 papers (14.7%) and a control group of 85,373 papers (30.0%). The treatment group is relatively smaller, because proportionally more COVID-19 research has been done by researchers outside the US, especially those in China and Italy.

As a robustness check, we selected a random sample of 300 publications from the treatment group and looked up information supplied by the publishers on submission and publication dates. Far from all publishers offer this information and to our knowledge there are no databases gathering this information consistently. Thus, we were able to find submission dates for 153 (51.0%) of the 300 publications. Of these, 129 (84.3%) were submitted after 15 March 2020, and 276 of the 300 (92.0%) were published after this date.

We used mixed logit models with random intercepts and random slopes to estimate the relationship between the dichotomous intervention variable (2019 sample = 0, COVID-19 sample = 1) and (i) women's share of first authorships (outcome variable: man = 0, woman = 1), (ii) women's share of last authorships (outcome variable: man = 0, woman = 1), and (iii) women's overall representation per article (two-vector outcome variable: number of women, number of men; *Crawley, 2012*). We included scientific journal as random effect parameter to adjust for variations in COVID-related research activities across scientific journals.

For the time factor analysis, we used the date of electronic publication (or date of publication, if electronic publication date was not available) from PubMed to create dichotomous variables for COVID-19 studies published in March/April 2020 and May 2020. Following the procedure specified above, we used mixed logit models to

estimate the relationship between these time-specific dichotomous variables and the three outcome measures.

The statistical analyses were conducted in R version 4.0.0. For the mixed logit models, we used the 'lme4' v. 1.1–23 package in R. We used the 'emmeans' v. 1.4.7 package to produce adjusted means and 'ggplot2' v. 3.3.0 to produce figures.

To produce *Table 1*, we manually categorized journals by specialty. Four authors participated in grouping the journals, with at least two independently coding every journal, and with discrepancies addressed by team consensus.

**Jens Peter Andersen** is in the Danish Centre for Studies in Research and Research Policy, Department of Political Science, Aarhus University, Aarhus, Denmark

https://orcid.org/0000-0003-2444-6210

**Mathias Wullum Nielsen** is in the Department of Sociology, University of Copenhagen, Copenhagen, Denmark

https://orcid.org/0000-0001-8759-7150

**Nicole L Simone** is in the Department of Radiation Oncology, Sidney Kimmel Cancer Center, Thomas Jefferson University, Philadelphia, United States

https://orcid.org/0000-0002-7662-7470

**Resa E Lewiss** is in the Department of Emergency Medicine, Thomas Jefferson University, Philadelphia, United States

https://orcid.org/0000-0002-9512-4342

**Reshma Jagsi** is in the Department of Radiation Oncology, University of Michigan, Ann Arbor, United States

rjagsi@med.umich.edu

https://orcid.org/0000-0001-6562-1228

*Author contributions:* Jens Peter Andersen, Mathias Wullum Nielsen, Conceptualization, Data curation, Software, Formal analysis, Validation, Investigation, Visualization, Methodology, Writing - original draft, Writing - review and editing; Nicole L Simone, Resa E Lewiss, Conceptualization, Validation, Investigation, Methodology, Writing - original draft, Writing - review and editing; Reshma Jagsi, Conceptualization, Supervision, Validation, Investigation, Methodology, Writing - original draft, Project administration, Writing - review and editing

*Competing interests:* Resa E Lewiss: Founder of TIME'S UP Healthcare, a non-profit initiative that advocates for safety and equity in healthcare; advisor for FeminEM.org, a website that supports the careers of women in medicine. Reshma Jagsi: Has stock options as compensation for her advisory board role in Equity Quotient, a company that evaluates culture in health care companies; has received personal fees from Amgen and Vizient and grants for unrelated work from

the National Institutes of Health, the Doris Duke Foundation, the Greenwall Foundation, the Komen Foundation, and Blue Cross Blue Shield of Michigan for the Michigan Radiation Oncology Quality Consortium; has a contract to conduct an investigator-initiated study with Genentech; has served as an expert witness for Sherinian and Hasso and Dressman Benzinger LaVelle; uncompensated founding member of TIME'S UP Healthcare; member of the Board of Directors of ASCO. The other authors declare that no competing interests exist.

## Funding

No external funding was received for this work.

### Decision letter and Author response

Decision letter https://doi.org/10.7554/eLife.58807.sa1
Author response https://doi.org/10.7554/eLife.58807.sa2

## Additional files

### Supplementary files

• Supplementary file 1. Details of the mixed regression models used to estimate the adjusted means and 95% confidence limits shown in *Figure 1*.

• Transparent reporting form

### Data availability

The final dataset for the main analysis is available on OSF: https://osf.io/cpv2m/.

The following dataset was generated:

| Author(s) | Year | Dataset URL | Database and Identifier |
|---|---|---|---|
| Andersen JP, Nielsen MW | 2020 | https://osf.io/cpv2m/ | Open Science Framework, cpv2m |

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
