## [Decision Letter]

Thank you for submitting your article "Meta-Research: Is Covid-19 Amplifying the Authorship Gender Gap in the Medical Literature?" for consideration by *eLife*. Your article has been reviewed by two peer reviewers, and the evaluation has been overseen by the *eLife* Features Editor (Peter Rodgers). The reviewers have opted to remain anonymous. Both reviewers were positive about the article, but they raised a small number of concerns that we would like you to address in a revised manuscript.

Summary:

Reviewer #1: This is an analysis of whether COVID-19 is affecting the gender gap in medical publishing. The authors identify papers published on COVID-19 in 2020 and analyze whether women are underrepresented as first author, last authors or general authorship by comparing their representation in the same journals for all of 2019. They find that women are underrepresented, most strongly in the first author category. Further, this disparity is unlikely to be explained by the underrepresentation of female authors in specific fields that characterize papers published on COVID-19. It's a very clever and convincing analysis but some details need to be clarified.

Essential revisions:

1) The full dataset used in the study (all papers, author names, gender inferences, etc), as well as the code used to analyze the data, need to be made available to allow others to reproduce or expand the analysis.

2) The robustness of author gender determination is not clear, and the methods should be described in more depth. For example, the text states "Gender was reliably estimated for 81.9% of the entire sample": what does "reliably estimated" mean here? How is the accuracy of the gender estimation tool established? What is the likelihood that a given gender was determined correctly? Doing spot checks of a random sample of authors can be helpful. Additionally, adding more information on the specific methods used for inferring gender from author names would be useful.

3) The same issue might lead to bias in the analysis: only 81.9% of authors are included in the study, and these 81.9% represent a biased sample of all authors - those with names that are more common or more recognizable by the gender estimation tool. Thus, the study likely under-represents authors from certain countries, nationalities, and social backgrounds. Is there a way to assess the impact of this bias on the results and analysis? At a minimum, this should be discussed as a caveat.

4) I am not convinced by the authors' argument that the observed differences are not due to a specialty bias. By looking at COVID-19-specific papers only, they are looking at the gender distribution of authors in a specific field, and comparing them to the gender distribution of authors in a variety of other fields who happen to publish in similar journals. It is entirely possible that there is a gender bias in coronavirus research, for example- this would not be captured by journal specialty, but would be seen in gender proportions for COVID-19 papers. Please add additional columns to table 1 looking at all research published in these journals in 2020 (not just COVID-19-related research), and/or discuss at greater length the possibility of specialty bias in your results.

5) The authors limited their samples to papers with at least one US author. It could be useful to know, of the 1179 papers in the treatment group, how many papers had more than one US author? I'm concerned that this limitation may introduce confounding factors. Knowing how many of these papers were published by teams of scientists primarily working in the US, vs a single scientist working in the US with a larger international team, would address this concern and more strongly support the authors' point that the disparity in female authorship can be potentially linked to the social restrictions in the US.

---

## [Author Response]

[We repeat the reviewers’ points here in italic, and include our replies point by point, as well as a description of the changes made, in Roman.]

We want to thank the editor and the reviewers for their excellent comments and suggestions. We have gone over them all and made changes accordingly. Most importantly, some of the comments have caused an update of the data set, which changes some of the findings and allows us to fine-tune the analysis. Since we initially harvested data in April 2020, PubMed has changed its interface and now provides better (and more consistent) information on author affiliations. This allows a much more precise analysis of author country, as requested by one of the reviewers. This was not possible previously.

We address this data update in the paper, in an effort to be as transparent about the process as possible, but for good measure, we would like to mention here the original sample, which was 9,050 COVID-19 papers, of which 1,179 had gender for first and last author, and at least one US-based author. These numbers are now 15,839 and 1,893 respectively. When analyzing first-authors, we restrict to US-based first authors and vice versa for last authors.

Below, we list the comments from the reviewers and editor in italics and our responses in regular font face.

Essential revisions:1) The full dataset used in the study (all papers, author names, gender inferences, etc), as well as the code used to analyze the data, need to be made available to allow others to reproduce or expand the analysis.

We have uploaded the data and the analytical scripts to OSF. The project is private right now, but we will make it public when/if our paper is published. An R script is provided, which will allow full analysis from the most raw data set we are able to provide (gender is inferred and information from PubMed is extracted into a table).

2) The robustness of author gender determination is not clear, and the methods should be described in more depth. For example, the text states "Gender was reliably estimated for 81.9% of the entire sample": what does "reliably estimated" mean here? How is the accuracy of the gender estimation tool established? What is the likelihood that a given gender was determined correctly? Doing spot checks of a random sample of authors can be helpful. Additionally, adding more information on the specific methods used for inferring gender from author names would be useful.

Checking the correctness of gender assignment is of course important. Our data relies on an algorithm, which we have previously used and checked for robustness. We realize that this was not apparent from the very short comment in the paper, and have added a little more detail about which algorithm we used and that robustness checks have previously been performed.

3) The same issue might lead to bias in the analysis: only 81.9% of authors are included in the study, and these 81.9% represent a biased sample of all authors - those with names that are more common or more recognizable by the gender estimation tool. Thus, the study likely under-represents authors from certain countries, nationalities, and social backgrounds. Is there a way to assess the impact of this bias on the results and analysis? At a minimum, this should be discussed as a caveat.

We have adjusted the method so that we now only compare US first and last authors. We believe this is a strength of the study because it allows us to isolate the effect to a single country and the timing of both disease and social constraints. For the analysis of the full author group, there could potentially be a bias, which we address in the limitations.

4) I am not convinced by the authors' argument that the observed differences are not due to a specialty bias. By looking at COVID-19-specific papers only, they are looking at the gender distribution of authors in a specific field, and comparing them to the gender distribution of authors in a variety of other fields who happen to publish in similar journals. It is entirely possible that there is a gender bias in coronavirus research, for example- this would not be captured by journal specialty, but would be seen in gender proportions for COVID-19 papers. Please add additional columns to table 1 looking at all research published in these journals in 2020 (not just COVID-19-related research), and/or discuss at greater length the possibility of specialty bias in your results.

While this is certainly an important point, we also have to disagree, for two reasons. COVID-19 specific papers are not from a specific field. On the contrary, the vast majority of medical specialties have picked up research on COVID-19 in one way or another. The search for cures or vaccines are likely limited, but discussions about implications for treatment of other diseases span all of clinical medicine. Secondly, our reason for using COVID-19 research as the “treatment” case was not to look specifically into this, but rather to have a sample of papers where we knew the research would have to be done in 2020. If we introduced a new column of “other” 2020 papers from the same journals, it would be impossible to know which of these had been in the pipeline since 2019 or earlier, and which were actually based on research from 2020.

5) The authors limited their samples to papers with at least one US author. It could be useful to know, of the 1179 papers in the treatment group, how many papers had more than one US author? I'm concerned that this limitation may introduce confounding factors. Knowing how many of these papers were published by teams of scientists primarily working in the US, vs a single scientist working in the US with a larger international team, would address this concern and more strongly support the authors' point that the disparity in female authorship can be potentially linked to the social restrictions in the US.

We hope that the changes to how we analyze first and last authors (analysis of first/last author gender restricted to first/last authors from the US) also addresses this point. For the share of women in the full group, this remains a valid concern, but not one we are currently able to address. While PubMed has now introduced per-author affiliations, it is not realistic to use these to make statistical analyses. When we claim to look only at papers with US-authors, this is really a conservative claim; there are certainly more papers with US addresses, but where the country or state is not included in the affiliation, or included in non-standardized forms. Our database contains more than sixty thousand “countries” from the affiliations, as there is no standard formatting of PubMed affiliations. We have trawled all of these for combinations of “USA”, “United States”, “America”, state names, names of major cities etc., to find the most complete set of affiliations related to the USA. However, we are not able to provide a reliable estimate of the national distribution of affiliations of all authors.